# Should I Stay or Should I Go? Explaining the Turnover Intentions with Corporate Social Responsibility (CSR), Organizational Identification and Organizational Commitment

Erum Shaikh [1],*, Mohsen Brahmi [2], Pham Chien Thang [3], Waqas Ahmad Watto [4], Ta Thi Nguyet Trang [5] and Nguyen Thi Loan [6]

1   Department of Business Administration, Shaheed Benazir Bhutto University, Sanghar Campus, Sindh 68100, Pakistan
2   Faculty of Economics and Management, University of Sfax, Glacette Airport Sfax, Sfax 2100, Tunisia; brahmi.mohsen@gmail.com
3   Faculty of Journalism and Communication, TNU-University of Sciences, Thai Nguyen City 24000, Vietnam; thangpc@tnus.edu.vn
4   Department of Commerce, Bahauddin Zakariya University, Multan 60000, Pakistan; waqasbzu67@gail.com
5   Department of Economics and Management, TNU-International School, Thai Nguyen City 24000, Vietnam; trangttn@tnu.edu.vn
6   Faculty of Economics and Business Administration, Hong Duc University, Thanh Hoa City 40100, Vietnam; nguyenloan@hdu.edu.vn
*   Correspondence: erumshaikh0@gmail.com

**Abstract:** The aim of this paper was to investigate the relationship between corporate social responsibility (CSR), organizational commitment, and organizational identification with turnover intentions. This paper also explains the mediating relationship between organizational commitment and organizational identification with the corporate social responsibility and turnover intentions. The data were gathered from banking professionals working in different banks in Pakistan. The participants were recruited through convenient sampling; in total, three hundred participants were involved in this study. The resulting data were analyzed, and the conclusions were drawn through regression and correlation analysis using the SPSS Software. The findings of this study show that corporate social responsibility plays a significant role in determining the organizational commitment and organizational identification of internal stakeholders and employees in financial institutions. This study will be help organizations determine their social responsibility and all the benefits that they can receive through the implementation of CSR practices.

**Keywords:** CSR; turnover intentions; organizational identification; organizational commitment; social identity theory (SIT)

## 1. Introduction

Corporate social responsibility (CSR) plays a very major role in evaluating the performance of organizations, as stated by many stakeholders globally. Currently, it is considered obligatory that organizations should fulfill their social responsibility in accordance with their economic objectives. Organizations should practice corporate social responsibility activities to gain different stakeholder benefits, such as enhancing their reputation by contributing to the development of society and increasing their performance by valuating internal and external stakeholders.

Recently, the patterns of conducting business have changed. Organizations are focused not only on profit seeking, but they also consider the benefits of stakeholders as well [1–3]. Corporate social responsibility (CSR) focuses on worthy causes such as protecting the environment, and such business practices, which are socially beneficial, represent the business concerns regarding ethical issues on part of their organization [4]. In order to

build a strong relationship with all stakeholders, corporate social responsibility is an essential factor [5].

Stakeholder theory [6,7] states that the performance and efficiency of an organization depends upon the attitude and behaviors of many individuals and group-based stakeholders. It is believed that the claims and requirements of all stakeholders must be fulfilled in order to survive under today's tough business conditions and to ensure a long-term relationship with stakeholders [8].

Corporate social responsibility (CSR) is more concerned with the phenomenon of increasing corporate scandals [9]. Previous studies have demonstrated that all stakeholders gain benefits from CSR practices [10]. Despite the benefits, CSR has not been properly explored in Pakistan [11]. CSR offers competitive advantages [11] because it contributes towards the retention of skillful employees and building a corporate image [12]. When an organization has gained the reputation that it serves the society and employees feel better working for the organization. As a result, employees are more committed to the organization and their turnover intentions are decreased.

Research has shown that senior management teams always face problems caused by employees' turnover intentions [13]. Turnover intentions have two reasons, namely, conflicts with management and the availability of a better opportunity. Consequently, the cost of hiring and training employees increases on the part of the organization and the organization also loses a skillful and trained workforce. The performance and output of the organization decreases because of the employees' turnover intentions. Previous studies suggest different ways of decreasing employees' turnover intentions [14]. The authors [15] stated that there is a give and take relationship between the organization and its employees. This relationship is regularly disturbed within the organizational atmosphere and employees decide whether to remain in the organization. The authors in [16] stated that senior management teams must necessarily implement some procedures to keep employees in the organization, whether this is in the form of organizational commitment or organizational identification.

Organizational commitment (OC) defines the employee's affiliation to their organization [17]. Committed employees widely consider the organization's vision and mission statement and their contribution is more aimed towards their organization [18]. The author(s) in [19] stated that organizational commitment has a positive relationship with CSR. This study also explored the relationship with CSR and commitment in the context of social identity theory (SIT), which claims that individuals identify themselves with the organization in which they work. Therefore, CSR perception has positive results with regard to commitment [20,21].

Previous studies have mentioned that employees attribute extra value and recognition to those organizations which are socially responsible and ethically sound. Employees' loyalty and commitment increase and there is a sense of honor involved in becoming part of that organization. According to [22], ethical and social responsibility enhances organizational commitment with the exchange of relationship among employees. Social identity theory (SIT) is based on organizational identification [23,24]. Authors in [25] endorsed the fact that organizational identification contributes towards employee resilience divergence situation. The organizational mission can be achieved through organizational identification [26].

Research in [27] shows that members which greatly identified with the organization are more positive about and toward the organization. Organizationally identical employees believe in the organization that produces valuable outputs. CSR activities develop a distinctive image of organization which contributes to the organization identification [28]. The author in [29] suggests that organizational identification enhances organizational commitment, organizational citizenship behaviors, engagement, job satisfaction, and thus lowers turnover intentions [30]. According to [31], organizational identification has a significant relation with turnover intentions and employees remain committed with socially responsible organizations. The authors found a significant impact of CSR on the turnover

intentions of the employees with the mediating role of organizational identification and organizational commitment in banking sector of Pakistan.

Furthermore, study provides important insights to the HR mangers on how to reduce turnover intentions. Employees turn over intentions have been reduced with effective CSR practices, organizational commitment, and organizational identification. Ref. [32] findings suggest that turnover is main challenge which is addressed by the modern organizations. According to [33], turnover enhances workloads and demands on remaining employees so that there is extra pressure on employees, which causes burnout and a greater turnover.

This research aims to elaborate and answer following question:

1.  Does corporate social responsibility have any influence on employees of the organization?
2.  Does corporate social responsibility play any role in organizational commitment?
3.  Does corporate social responsibility play any significant role in organizational identification?
4.  Does corporate social responsibility have any impact on job satisfaction of employees in the organization?
5.  Is corporate social responsibility associated in any terms with turnover intentions of the employees?

## 2. Literature Review

Corporate Social Responsibility (CSR) takes care of employees' wellbeing because of the supportive leadership which strengthens the relationship of organization with their salespeople. Where there is more harmony between the values of employees and organization, there will be less intention of the employees to leave the organization [34]. Happy employees are more likely to be satisfied and committed, thus want to stay with the firm [35]. According to [36], CSR activities have a significant effect on the performance of the organizations which fascinate and hold the employees and they remain committed with their organization. However, previous literature was not able to find out the relation between the turnover intentions of the employees and the ethical climate of the organization [37]. Rather, authors highlighted that the CSR has indirect effect on the turnover intentions when the employees are satisfied [38,39] and commitment with organization [40].

CSR initiatives help organizations to identify themselves by instilling a sense of pride in their membership [41]. Employees are more likely to be satisfied with their current work relationships as a result of this; they do not waste their time to seek new employment anywhere else. Few studies have looked at the effect of perceptive CSR on employee intentions to resign.

The author [42] stated that employee involvement in CSR has negative relation with the intentions of employees to leave the job. Authors [43] suggested that perceived organizational trust mediates and has a positive and direct relationship between CSR and loyalty intention. They [44] did an empirical study on Dutch ABN-AMRO bank where the employees of the bank performed various CSR programs voluntarily. The authors found that there are not any differences in attitude between the groups (program participants, community volunteers and non-volunteers) when it comes to the importance of their personal career, they are very much committed with their organization, towards the intention to leave their job the level of their Organizational Citizenship Behavior (OCB).

According to the research of [45,46], CSR activities create identification of the organization which will affects positively on the intention of the employees to stay in the organization. Although there was a positive support in mediation between the intention of the employees to stay and OCB loyalty, there was not any evidence for mediation on the behavior of the supervisor reported. Given the paucity of empirical data and contradictory outcomes, further information on whether workers' perceptions of CSR impact their desire to resign from job. In particular, authors hypothesize that:

**Hypothesis 1.** *Corporate Social Responsibility (CSR) relates negatively to Turnover Intentions (TI).*

Previous studies suggest that organizational commitment is positively and significantly associated with corporate social responsibility. In a similar vein, [47] argued that the CSR practices of organizations increase the organizational commitment also increases. Similarly, authors [48,49] conducted research in the financial sector. Findings suggest that OC and CSR are positively related with each other. Additionally, the results from [50] suggest that organization identification mediate the relationship between organization commitment and CSR. Authors in [51] state that "CSR is positively nexus with OC".

Based on the framework of SIT, many recent research works explore the impact of CSR on the commitment of the employees. It also explores the relation of explicit CSR with the attitude of the employees. People in the sociocultural settings prefer to classify themselves into distinct social groups based on SIT [52]. People frequently use to compare and evaluate themselves with their organizations to others to form or improve a good self-image self. Unfavorable contrasts might harm their self-esteem, whereas positive contrasts support their self-concept about organizations [53,54]. Hence, employees feel proud when they are associated with the positive collective honors [55].

According to the theory of met expectations [56], individuals are more likely to stay in an organization if their growth requirements are satisfied. [57] SIT demonstrates the positive relation of CSR with OC. They backed up their predictions with substantial data from a financial services employee firm. Authors confirmed in [58] that workers' attitudes on CSR are positively connected to OC by applying organizational justice theory. Besides, in [59] through applying the organizational identification process, authors studied the perception of the employees about CSR, leading to an improvement in organizational identity. Additionally, if individuals are dedicated to their jobs, they will look for ways to advance their careers inside their present organization. Therefore, on the basis of the above literature rationales, authors proposed that:

**Hypothesis 2.** *Corporate Social Responsibility (CSR) relates positively to Organizational Commitment (OC).*

Current studies on micro-CSR recommended that CSR has great influence on organizational identification of the employees [60–62]. When the organizations are involved and performing activities for social welfare, employees feel their strong association with organization [63]. Additionally, [64] stated that employees feel very comfortable and committed when their employers support them in performing the community-based activities. The study referenced in [65] claims that the organizations who are actively engaged in CSR activities for social wellbeing have a favorable reputation and are better equipped to recruit qualified personnel. The author of [66] states that external and internal stakeholders are more likely to identify with a company if they are aware of its CSR initiatives.

**Hypothesis 3.** *Corporate Social Responsibility (CSR) relates positively to Organizational Identification (OI).*

Value fit commitment can also decrease the rate of intention to leave the job. Organizational commitment is defined as it is linked with a strong desire amongst employees to remain within their organizations [67,68]. Commitment of the employees should be driven by their intents to either leave or stay in an organization when they work for unethical organizations or by the enhanced the equality that they perceive when they work for socially responsible organizations. This suggested link has been supported by previous studies. For example, organizational commitment and Person-Organizational Fit (P-O fit) these both variables are positively and reliably associated with the turnover intentions of the employees [69,70].

Commitment of the employees and job satisfaction of the employees [71], and Person-Organizational Fit, P-O fit [72] all are intent to leave the job. Furthermore, earlier research has shown that when a P-O fit is poor that will cause turnover intentions [73,74]. Person-Organizational Fit as per [75] was found to be associated with the emotional commitment,

intention to stay in the organization and other attitudes of the employees. It has also been proposed that P-O fit based on ethics or the alignment of private and corporate ethics, is linked with the commitment level, job satisfaction, turnover intentions, and desirability of employees to be within the organization [76–78]. On the basis of the above literature, the following hypothesis is presented:

**Hypothesis 4.** *Organizational Commitment (OC) relates negatively to Turnover Intentions (TI).*

Employees identify organizations with their actions in accordance with its norms and values [79,80]. Organizations have many norms and core values but one of them is employee's retention [81]. Organization identification has been associated with employees of the organization; therefore, they stay with organization [82]. Strong identification with an organization means that employees feel that the organization is part of them and are psychologically linked with the organization. Organization identification contributes toward self-enhancement, self-continuity, and lower uncertainty [83–85]. Self-image of the employees should be incorporated with characteristics of the organization as employees are identified with organization.

Turnover intentions of the employees feel the loss part of my self [86]. Consequently, turnover intentions decrease as identification increases. Likewise, many studies have shown that organizational commitment has a positive effect on the turnover intentions of the employees. [87–89] discussed that the organizational commitment of the employees can be increased with the initiatives taken with internal marketing, which has a great impact on performance and satisfaction of the job and turnover intention. The literature suggests that turnover intension is very important outcome of the organizational commitment of the employees [90]. The authors of [91] claim that when employees are highly committed with the organization, they have fewer tendencies to leave their job. As per [92], organizational commitment has a negative effect on the turnover intensions of the employees and positive relation with job satisfaction. Therefore, on the basis of the above literature rationales, the authors presented following hypothesis:

**Hypothesis 5.** *Organizational Identification (OI) relates negatively to turnover Intentions (TI).*

Recent literature has also shown that an organizational CSR can have a positive and direct relation with the commitment of the employees with the organization [93–95], stated that high organizational commitment can decrease the turnover intentions of employees [96,97]. Therefore, the authors presented following hypothesis:

**Hypothesis 6.** *Organizational Commitment (OC) mediates between Corporate Social Responsibility (CSR) and turnover Intentions (TI).*

According to [98], CSR can promote the external evaluation of the status and attractiveness of the organization, which provokes the desire of the employees to remain associated with the current organization and show strong commitment which highlights the strong values and sense of belongingness with the organization resulting in great satisfaction among the employees for their organization. People have great urge to discover a positive image of social group, which helps them to better enhance their own self-concept and self-image [99]. Meanwhile, workers are more likely to connect with those organizations which they believe are renowned and have positive status, which might help them to increase their self-esteem and positive self-image [100].

According to the previous literature, CSR is closely linked to societal assessments of regard and esteem with which firms are viewed in society [101–103]. CSR have a great impact on the corporate perceptions and product outcomes [104–106]. Organizations, which are more responsible towards the society, are more appealing to potential workers [107,108]. Being socially responsible organizations can enhance the image of the organization in

comparison to other organizations which are not responsible towards society. On the basis of the above literature, authors presented following hypothesis:

**Hypothesis 7.** *Organizational Identification (OI) mediates between Corporate Social Responsibility (CSR) and Turnover Intentions (TI).*

### 3. Materials and Methods

This is descriptive and explanatory study which complains the phenomenon or situation; thus, this study focuses on the current situation rather than focusing decision making. This type of study always focused the developed hypothesis in terms of verification and its objectives to prove that stated hypothesis.

The purpose of this study is to find the relationship between corporate social responsibility and turnover intentions in connection with organizational commitment and organizational identification. The population of the study consists of banking sector employees. CSR has a great importance for the wellbeing of the employees; supportive leadership can improve the relation of employees with the organizations, and employees have less intension to leave the organization [109].

The target population in this study is banking employees working in different banks of Lahore, Pakistan. The data are gathered from the employees of five banks working in different branches in Lahore. These five banks are chosen because these banks are the very oldest banks—they have branches across the country and in foreign countries. The reason of selecting this population is easy approach and management of short time. It is because the data can be gathered easily from bank employees. These all banks are private banks, and no bank is chosen from the public sector. In order to be more focused, this study only collected data from the banks of Lahore region. Lahore is very famous city for business, tourism, culture, education, and entertainment. These five banks from the private sector were selected, and the study confirmed that the Lahore region is unique, which can give interesting findings and new additions in the latent literature. Thus, the data were obtained from the employees of Habib Bank Limited (HBL), Allied Bank Limited (ABL), United Bank Limited (UBL), Muslim Commercial Bank (MCB) and NDLC-IFIC Bank (NIB). Sample size adequacy is very important issue in research work. The research detailed in [110] suggested that constructing items measure from 3 to 4, and small respondents are satisfactory when the reliability is <70. On the other hand, [111] supports the idea that a sample size of 50 to 100 is poor, 200 to 300 is good, and 500 to 1000 is excellent. A sample size was 300 collected for this study from the employees of different banks with the help of convenient sampling. Convenient sampling is a nonprobability sampling technique and is used when the authors have limited resources such as limited time and workforce. Furthermore, it has geographical proximity, easy accessibility, affordable and willingness to participate by the respondents. For descriptive analysis, frequency, percentage, data normality and statistical analysis such as reliability, correlation, and regression analysis are applied to understand the data, draw conclusions, and conclude the results.

This study was conducted in Lahore city, where the total 450 questionnaires were distributed to the employees of banks, from which 300 questionnaires were received back and used for analysis. This research study suggests that there is negative relationship between corporate social responsibility and employee's turnover intentions. When employees are motivated and appreciated within the organizations then they become committed and loyal to their jobs and organizations [112]; hence, corporate social responsibility enhances organizational commitment of the employees.

This research model investigates the relationship between Corporate Social Responsibility (CSR), Organizational Commitment (OC); Organizational Identification (OI) with Turnover Intentions (TI). TI also explains the mediating relationship of Organizational Commitment and Organizational Identification between Corporate Social Responsibility and Turnover Intentions.

*Framework*

Figure 1 shows the framework of this study which determines the corporate social responsibility which appears in the further sub classification of organizational commitment, organizational identification, and jobs satisfaction. These aspects influence the turnover intention of the management to adopt CSR for the growth of the organization, as well as the development of the employees.

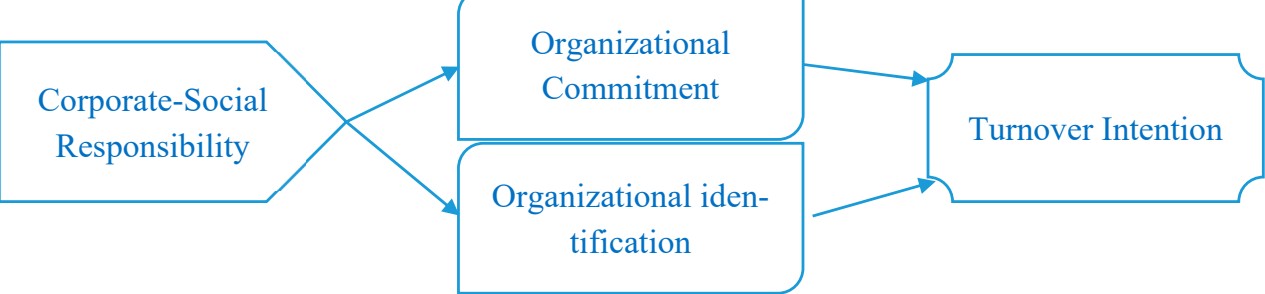

**Figure 1.** Framework.

## 4. Results

The reliability of data in Table 1 and latent variables are measured with Cronbach's alpha values. The recommended criteria value for Cronbach's alpha ≥0.70 [113]. Cronbach's alpha value for organizational commitment and organizational identification is 0.70, corporate social responsibility and turnover intentions are 0.84 and 0.82. The overall reliability of the questionnaire is 0.790.

**Table 1.** Reliability Statistics.

| Variable | No. of Questions | Cronbach's Alpha |
|---|---|---|
| Organizational Commitment | 1–8 | 0.715 |
| Organizational Identification | 9–14 | 0.708 |
| Corporate Social Responsibility | 14–20 | 0.847 |
| Turnover Intentions | 20–25 | 0.826 |
| Overall Reliability | 1–25 | 0.790 |

Source: this study (own contribution).

Table 2 shows the results of correlation coefficient which describes the relationship of one variable with another variable. Its value is between +1, 0, −1. Correlation coefficient value +1 positive relationship and −1 for negative relationship and 0 mean there is no relationship between the variable. Corporate social responsibility has a highly significant positive relationship with Organizational Commitment (0.536 **), $p < 0.01$; therefore, it supports our hypothesis that CSR plays a significant role in influencing organizational commitment. CSR has a positive correlation with organizational identification, (0.645 **), $p < 0.01$; therefore, it supports our hypothesis that CSR has a positive influence on organizational identification. CSR has a negative influence on turnover intentions, (0.407 **), $p < 0.01$; therefore, it supports our hypothesis that CSR has a negative influence on turnover intentions. Organizational commitment has a highly significant negative relationship with turnover intentions (−0.157 **), $p < 0.01$; therefore, it supports our hypothesis that organizational commitment has a negative influence on turnover intentions. Organizational identification has a highly significant negative relationship with turnover intentions (−0.157 **), $p < 0.01$; therefore, it supports the hypothesis that organizational identification has a negative influence on turnover intentions.

**Table 2.** Correlation Table.

|  | Corporate Social Responsibility | Organizational Commitment | Organizational Identification | Turnover Intentions |
|---|---|---|---|---|
| Corporate Social Responsibility | 1 |  |  |  |
| Organizational Commitment | 0.536 ** | 1 |  |  |
| Organizational Identification | 0.645 ** | 0.297 ** | 1 |  |
| Turnover Intentions | −0.407 ** | −0.157 * | −0.308 ** | 1 |

** Correlation is significant at the 0.01 level (2-tailed). * Correlation is Significant at the 0.05 level. Source: This Study.

### 4.1. Regression Results

Table 3 shows that the beta results and beta value are −0.932, which means that change in CSR also changes the turnover intentions of the employees. F value is 169.131 > 15, so model is good fit. $R^2$ is over all explanatory power of the model which is 0.362 that the 36.2% and adjusted $R^2$ is 0.360 that means there is a 36% variation in turnover intentions due to corporate social responsibility. The level of significance is 0.000 < 0.05; therefore, H1 is accepted in this study.

**Table 3.** CSR with turnover intentions.

| Variable | B | R Square | Adjusted R Square | T | F | Sig. |
|---|---|---|---|---|---|---|
| Corporate Social Responsibility | −0.932 | 0.362 | 0.360 | 13.005 | 169.131 | 0.000 |

Source: This Study.

In Table 4, the beta value is 0.126, which means that change in CSR also changes the organizational commitment. The F value is 34.33 > 15; thus, the model is a good fit. $R^2$ is the overall explanatory power of the model, which is 0.77; the adjusted $R^2$ is 0.75, which means that there is a 75% variation in the organizational commitment due to corporate social responsibility. The level of significance is 0.000 < 0.05; therefore, the H2 is accepted.

**Table 4.** CSR regress on organizational commitment.

| Variable | B | R Square | Adjusted R Square | T | F | Sig. |
|---|---|---|---|---|---|---|
| Corporate Social Responsibility | 0.126 | 0.77 | 0.75 | 5.860 | 34.33 | 0.000 |

Source: this study.

In Table 5, the beta value is 0.121, which means that change in CSR also changes the organizational identification. The F value is 69.971 > 15; therefore, the model is a good fit. $R^2$ is the overall explanatory power of the model, which is 0.62; and the adjusted $R^2$ is 0.60, which means that there is a 60% variation in organizational identification due to corporate social responsibility. The level of significance is 0.000 < 0.05; therefore, the H3 is accepted.

**Table 5.** CSR regress on organizational identification.

| Variable | B | R Square | Adjusted R Square | T | F | Sig. |
|---|---|---|---|---|---|---|
| Corporate Social Responsibility | 0.121 | 0.62 | 0.60 | 6.305 | 69.971 | 0.000 |

Source: this study.

In Table 6, the beta value is (−0.428), which means that change in organizational commitment also changes the turnover intention. The F value is 46.995 > 15; therefore, the model is a good fit. $R^2$ is the overall explanatory power of the model, which is 0.136; the adjusted $R^2$ is 0.133, which means there is a 13.3% variation in turnover intentions due to organizational commitment. The level of significance is 0.000 < 0.05; therefore, the H4 is accepted.

**Table 6.** Organizational commitments regress on turnover intentions.

| Variable | B | R Square | Adjusted R Square | T | F | Sig. |
|---|---|---|---|---|---|---|
| Corporate Social Responsibility | −0.428 | 0.136 | 0.133 | 6.855 | 46.995 | 0.000 |

Source: this study.

In Table 7, the beta value is (−0.554), which means that change in organizational identification also changes the turnover intentions. The F value is 47.995 > 15; therefore, the model is a good fit. $R^2$ is the overall explanatory power of the model, which is 0.49; the adjusted $R^2$ is 0.493, which means that there is a 49.3% variation in turnover intentions due to organizational identification. The level of significance is 0.000 < 0.05; therefore, the H5 is accepted.

**Table 7.** Organizational identification regresses on turnover intentions.

| Variable | B | R Square | Adjusted R Square | T | F | Sig. |
|---|---|---|---|---|---|---|
| Corporate Social Responsibility | −0.554 | 0.498 | 0.498 | 6.855 | 47.995 | 0.000 |

Source: this study.

### 4.2. Mediation Analysis

Table 8 shows the mediation effect of organizational commitment on corporate social responsibility and turnover intentions. In Model 1, all the control variables (demographics) are included and in Model 2 includes all the control variables, such as independent variables of the study, i.e., CSR regressed. In Model 3, all the control variables include independent variables, i.e., CSR and mediating variable OC regressed to measure the mediation effect. The statistics show that the beta value of the corporate social responsibility in Model 2 is 0.914 **, the beta value of CSR is 0.962 ** (0.914 ** > 0.962 **), and the beta value of organizational commitment is 0.244 *. It is less than the beta value of 0.962 ** (0.244 * < 0.962 **), which means that organizational commitment partially mediates the relationship between corporate social responsibility and turnover intentions.

**Table 8.** CSR, OC and TI.

| Variables | Model 1 | Model 2 | Model 3 |
|---|---|---|---|
| Gender | 0.065 | 0.052 | 0.049 |
| Education | 0.002 | −0.002 | −0.003 |
| Marital Status | −0.221 * | −0.0156 * | −0.156 * |
| Sector | 0.033 | −0.020 | −0.020 |
| Corporate Social Responsibility | | 0.914 ** | 0.962 ** |
| Organizational Commitment | | - | 0.244 * |

Dependent variable: turnover intention. ** $p < 0.05$, * $p < 0.01$. Source: this study.

The Table 9 shows the mediation effect of organizational identification on corporate social responsibility and turnover intentions. In Model 1, all the control variables include demographics of the study. In Model 2, all the control variables include independent variable, i.e., CSR regressed. In Model 3, all the control variables include independent

variable, i.e., CSR and mediating variables are organizational identification regressed to measure the mediation effect. The statistics show that the beta value of the corporate social responsibility in Model 2 is 0.914 **, the beta value of corporate social responsibility is 0.962 ** (0.914 ** < 0.777 **), and the beta value of organizational identification is 0.144 *, which is less than the beta value of 0.777 ** (0.144 * < 0.777 **), meaning that organizational identification partially mediates the relationship between corporate social responsibility and turnover intentions.

**Table 9.** CSR, OC, and OI.

| Variables | Model 1 | Model 2 | Model 3 |
|---|---|---|---|
| Gender | 0.065 | 0.052 | 0.045 |
| Education | 0.002 | −0.002 | −0.002 |
| Marital Status | −0.221 * | −0.0156 * | −0.155 * |
| Sector | 0.033 | −0.020 | −0.020 |
| Corporate Social Responsibility | | 0.914 ** | 0.777 ** |
| Organizational Commitment | | - | 0.144 * |

Dependent variable: turnover intention. ** $p < 0.05$. * $p < 0.01$. Source: this study.

Table 10 shows the list of the variables which are accepted/supported or partially mediated in this research study. Table 10 shows that hypotheses 1, 2, 3, 4, 5 are accepted/supported in this study and hypotheses 6 and 7 are partially mediated in this research study.

**Table 10.** Summary of Results.

| Hypotheses | | Result |
|---|---|---|
| 1. | Corporate social responsibility relates negatively to turnover intentions | **Accepted/Supported** |
| 2. | Corporate social responsibility relates positively to organizational commitment | **Accepted/Supported** |
| 3. | Corporate social responsibility relates positively to organizational identification | **Accepted/Supported** |
| 4. | Organizational commitment relates negatively to turnover intentions. | **Accepted/Supported** |
| 5. | Organizational identification relates negatively to turnover intentions. | **Accepted/Supported** |
| 6. | Organizational commitment mediates between corporate social responsibility and turnover intentions | **Partially Mediated** |
| 7. | Organizational identification mediates between corporate social responsibility and turnover intentions | **Partially Mediated** |

## 5. Discussion

The society is main stakeholder for banks and their operations effects on society [114]. CSR policies are return to society after the profitable operation of organization. When an organization implements CSR policies, the employee's organization identification is increased due to social exchange theory [115]. CSR and OC as well as OI are positively associated and have significant relations with each other. The authors of [116] claim that the banks who perform and start CSR initiatives for the welfare of the employees and society have more identification and favorable reputation with well-equipped and skilled workers. Conversely, the results of this study show that organizational commitment and organizational identification provide partial mediation between CSR and employee's turnover intentions [117]. According to this study, corporate social responsibility possesses significance in determining the organizational commitment of internal stakeholders banking

employees and organizational identification in relation to reduction in turn over intentions of financial sector employees [118].

This research has great implications. It shows that corporate social responsibility has much importance for the banks. Corporate social responsibility has its own significance in determining organizational commitment of the internal stakeholders that are its employees.

This study will guide the banks and other organizations in determining their social responsibility and all the benefits which they can receive through the implementation of corporate social responsibility practices. This research is helpful for the organizations, banks, corporate sectors as well as the internal and external stakeholders of the organization. Furthermore, this work is very helpful for the banks and organizations of developing countries because they have lack of technological resources and limited importance of the CSR practices.

Implementation of CSR practices are important for motivation, retention and future performance of the banking and non-banking employees. When organizations provide a sense of comfortable zone to their employees then they feel association and commitment with that workplace. Banks play a very vital role in society where they invest and earn. Therefore, when any firm starts practicing for the welfare (within and outside) of the organization, the commitment of employees and reputation is increased and employees feel more responsible, and their turnover intention is reduced. Organizational identification decreases the employees' intentions to quit their job when they themselves are considered important to the organization through CSR initiatives.

## 6. Conclusions

In this study, the relationship between corporate social responsibility and turnover intentions is studied in connection with organizational commitment, organizational identification, and job satisfaction as a mediating model. The results of this study explain that there is a positive relationship among corporate social responsibility and organizational commitment and organizational identification. It also explains that there is negative relationship between corporate social responsibility and turnover intentions. All the hypotheses of this study are supported herein. The findings of this study are very helpful for banking organizations of Pakistan. This study suggests corporate social responsibility has its own importance for organizations in determining the organizational commitment of internal stakeholders (employees) and organizational identification in relation to reduction in turnover intentions of the employees.

## 7. Limitation and Recommendations

This study was conducted with the employees of the banks only; thus, it is recommended that the same study may be carried out within other sectors such as telecom, manufacturing, healthcare, sugar mills, etc. Furthermore, the survey instrument was questionnaires, and sometimes the culture of the organizations and their complete dimensions related to the employees' conditions are not obtained by the use of questionnaires. Therefore, interviews may be used to obtain reliable results. This study only focused on the internal stakeholders of the organization that were employees of the organization; external stakeholders of the organization should also be focused on for more and better findings about the CSR.

**Author Contributions:** Conceptualization, E.S., W.A.W., Methodology, E.S., M.B., W.A.W.; Formal analysis, E.S., M.B., W.A.W., Investigation, E.S., W.A.W., P.C.T., T.T.N.T., N.T.L., Writing—original draft preparation, E.S., M.B., W.A.W., Writing—review and editing, M.B. All authors have read and agreed to the published version of the manuscript.

**Funding:** This paper does not receive any funding from any agency.

**Informed Consent Statement:** Not applicable.

**Data Availability Statement:** The supportive data will be provided on responsible request.

**Acknowledgments:** Authors are thankful to the Assigned Editor to handle the manuscript carefully and coordinate, cooperate and guide very kindly and timely. Furthermore, Authors are very thankful to Muhammad Nawaz Tunio to proof- read the article and assist to bring the paper in much better position.

**Conflicts of Interest:** There is no conflict of interest between any authors of this paper.

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
