# Peer review of "Should I Stay or Should I Go? Explaining the Turnover Intentions with Corporate Social Responsibility (CSR), Organizational Identification and Organizational Commitment"

_sustainability, doi:10.3390/su14106030_

Round 1

Reviewer 1 Report

Dear Author/s

Thank you for the opportunity to read your well-written study on concerning the turnover intentions with corporate social responsibility, organizational identification and organizational commitment. The proposed objective is interesting and ambitious.  The topic is timely and the authors add new value. In my opinion, the consideration of the mediating relationship is a great advantage of research. There is a need for papers that propose a new approach to exploring knowledge in this area and map possibilities for the future. The research approach proposed in this paper is undoubtedly in line with the latest research trends.

The structure of the work is an example of developing the subject of interest by the Author/s.

However, in its current stage, the manuscript can still be improved to be more valuable to the readers. Therefore, I encourage you to make minor changes that I believe will be made this paper stronger. Please see my detailed comments below. Good luck with your research!

I have provided my comments as follows:

Title – informative, reflecting the content of the paper, encouraging the reader to read the whole paper.

Abstract – complete

After reading a title and abstract, it is clear to me what the purpose of this article is, how the research was designed and what the main findings of the researchers are. Abstract is very informative.

Introduction

Correctly structured, giving the reader an introduction to the subject under study. The authors refer to current theories and research findings in CSR, organisational commitment and organisational identification. The introduction concludes with research questions. However, their formulation is in my opinion too obvious. In my opinion, the article would be stronger if the research questions were reformulated.  I encourage you to consider changing the sentences "have any influence, play any role, have any impact ..."

Literature Review

The researcher (s) draw from a variety of academic studies to make or support the points raised in the study. The literature review compliments the research conducted in that it gives the reader information from previous scholars in this field to consider as the reader reviews the current study.

Correctly structured, clearly explaining the hypotheses adopted and the conceptual framework of the study. In my opinion, this is an example of a sound scientific explanation of the approach to solving the problem adopted. In my opinion, the hypotheses are correctly formulated (supported by previous research) and correspond to the aim of the study.

Materials and Methods

In my opinion, this section would be stronger if the author explained the following:

In my opinion, the rationale for selecting the sample size is unclear (why was the data collected from 5 banks? it is a lot or a little?; why only in Lahere?). As the authors point out, the selection of the sample is a very important issue. I would encourage you to improve this section so that other researchers have the opportunity to repeat this study (e. g. in other countries). I believe that this would be very valuable for further studies on this subject.

Results - prepared correctly

Results are organized in a way that is easy to understand. In my view, this is an example of a clear and understandable presentation of research results. I find the results presented in the 'Mediation Analysis' sub-section particularly interesting. This is a relatively new extension of the analyses made in the research on the issues addressed.

Discussion and Conclusion of the Study  

In my view, these are conclusions rather than a discussion. The 'discussion' section should have confronted the findings with what has already been established in research on the subject. I did not find it in this section. I encourage you to expand this section or consider separating the "Discussion" and "Conclusions" sections.

References - prepared correctly

The paper draws on very recent literature. References include almost 100 items: in the case of a scientific article this is, in my opinion, an additional confirmation of a very solid literature review and of the author's preparation for the research.

It was a pleasure to read this manuscript. There is a lot of potential in this paper to make an interesting contrubution to the field. I hope you find the above observations useful as you continue to further develop your study. Good luck with your paper!

Reviewer 2 Report

I expect you will add many more valuable comments from the literature on Figure 1.

According to the author, this research study provides an answer to the following question: does corporate social responsibility affect the employees of an organization? Does corporate social responsibility play any role in organizational commitment? Does corporate social responsibility play any role in organizational identification? Does corporate social responsibility affect the job satisfaction of employees in the organization? Is corporate social responsibility in any way related to employees' intention to leave the organization?

These are well known questions and the answer to them is also trivial and obvious. 

Data was obtained from banking employees working in different banks of Pakistan. The sample size was 300 and was selected using convenience sampling. This is the advantage of this study - original data has been collected and analyzed.

Reviewer 3 Report

Dear authors,

I have read your paper with great interest. Unfortunately, despite the high relevance of the issue approached and the research efforts, the overall quality of your paper is seriously affected by the poor level of the English language that made quite difficult the process of assessing your paper.  

Nevertheless, I will present bellow a few reasons for which I would recommend the Editors to reject your paper:

  1. First of all, the paper incorporates some incorrect phrases from the very beginning. For instance in the Abstract section, one can find the following wording: “For descriptive analysis (frequency, percentage, data normality, and statistical analysis such as reliability, correlation, and Regression analysis is applied to conclude the results”. The sentence appears to lack its predicate, while the bracket that followed the phrase “descriptive analysis” was not closed afterwards. Besides, there are plenty of examples of incorrectly translated sentences or phrases throughout the content of the paper.
  2. There are several major flaws within the “Material and Methods” section. For instance, few insights are given in respect with the sampling method employed. Instead of providing valuable information regarding the sampling techniques, a brief reference regarding the total number of questionnaires disseminated and the number of responses received are presented in the ”Discussion and Conclusion of the Study” section (lines 438-439).
  3. The design of the methodology used in the paper is also missing from the “Material and Methods” section, as well as relevant information on statistical analyses conducted. For instance, Model 1, Model 2 and Model 3, which appear in table 4.8 and 4.9 are not properly substantiated within this important part of the paper. Furthermore, this section lacks in consistency as it includes a single subsection – “3.1. Framework”. This subsection contains only one figure without any supplementary developments. The text included in the first text-box in figure 3.1 is not entirely readable. Furthermore, despite mentioning twice the descriptive analyses performed by authors (see lines 22-24 and 262-265), no data are presented in this respect in the “Results” section or in any other part of the manuscript.
  4. Under the circumstances, the validity of the statistical determinations included in the paper is heavily affected by the shortcomings in terms of English language. Moreover, I consider that the flow of ideas in the manuscript should considerably be improved.

For the future, I would advise the authors to find a native English speaker, in order to proofread their manuscripts. Besides, I also advise them to pay closer attention to the manner they draw up and structure their manuscripts, so that the laborious statistical researches they are going to carry out further might be properly put to good use through publishing in relevant journals.

Round 2

Reviewer 3 Report

Dear authors,

I have read your revised version of the manuscript with a lot of receptiveness. Although a series of improvements have been brought to the initial version of your work, targeting a clearer explanation of the statistic determinations carried out and of the methodology employed, nonetheless, the work still contains, in my opinion, too many grammar mistakes that prevent it from being published in a high-impact journal of the international scientific community such as Sustainability. These errors regard the following aspects:

  • Wrong spelling of a series of words – for instance, a key word in your research is “Turnover Intentions”, while the phrase “Turnover Intensions” occurs approximately 11 times within the content of the manuscript (see lines 105, 127, 132, 133, 141, 143, 151, 229, 232, 272, 573);
  • There are frequent disagreements between the subject and the predicate of sentences;
  • A series of sentences lack their predicate;
  • There are sentences/phrases whose content is incomprehensible.

Consequently, although, based on a series of data gathered from a large sample of employees in the banking industry in Pakistan, you might have been able to bring important contributions to the development of the knowledge regarding the nexus between the corporate social responsibility, organizational identification and organizational commitment, I strongly believe that the importance of scientific writing should be considered as at least equal to the emphasis associated to the statistics, especially in the case of a journal with such a strong reputation as Sustainability.

Under such circumstances, I would be glad if the editor offers you the chance to rewrite your paper in a scientific language that matches the requirements for such a high-level work, so that you are able to disseminate the results of your research to the international scientific community.  

Kind Regards!

Author Response

Dear editor,

Thank you so much for assisting and handling our paper, and providing feedback from the referee which helped us to improve at a good range. Here are responses to the referee and the attached revised paper with the addition of one new co-author who has contributed to the development of the final draft. See the attachment. 

Best, Dr. Erum Shaikh.

No.

Comments from referee

Response from authors

1.      

I have read your revised version of the manuscript with a lot of receptiveness. Although a series of improvements have been brought to the initial version of your work, targeting a clearer explanation of the statistic determinations carried out and of the methodology employed, nonetheless, the work still contains, in my opinion, too many grammar mistakes that prevent it from being published in a high-impact journal of the international scientific community such as Sustainability. These errors regard the following aspects:

  • Wrong spelling of a series of words – for instance, a keyword in your research is “Turnover Intentions”, while the phrase “Turnover Intensions” occurs approximately 11 times within the content of the manuscript (see lines 105, 127, 132, 133, 141, 143, 151, 229, 232, 272, 573);
  • There are frequent disagreements between the subject and the predicate of sentences;
  • A series of sentences lack their predicate;
  • There are sentences/phrases whose content is incomprehensible.

Thank you so much for taking the time to read and review the draft and provide constructive feedback.

The article proofreads by an expert in the language and all errors and mistakes are rectified and corrected.  

2.      

Consequently, although, based on a series of data gathered from a large sample of employees in the banking industry in Pakistan, you might have been able to bring important contributions to the development of the knowledge regarding the nexus between the corporate social responsibility, organizational identification, and organizational commitment, I strongly believe that the importance of scientific writing should be considered as at least equal to the emphasis associated to the statistics, especially in the case of a journal with such a strong reputation as Sustainability.

Thank you for the kind suggestion, as there has already poured a lot with respect to the context in order to develop the knowledge, furthermore, the addition may change the perspective of the paper because we have to consider the aim and scope of the sustainability and its paper format.

3.      

Under such circumstances, I would be glad if the editor offers you the chance to rewrite your paper in a scientific language that matches the requirements for such a high-level work so that you are able to disseminate the results of your research to the international scientific community.  

Once again, many thanks for the kind suggestion, revised version of the paper is much improved.

Round 3

Reviewer 3 Report

Dear authors,

Again, I have read the second revision of your manuscript with high interest. While the explanations regarding the material and methods have been improved, I believe that the manuscript still needs some adjustments in order to be ready for publication.  Please take into consideration the following suggestions:

  1. Despite the fact that you have made several efforts to eliminate some grammatical errors, there are still quite a few parts of your article that need to be corrected in order to meet requirements demanded by high-impact scientific journals. Once again I find myself in the position of kindly reiterating the recommendation that I mentioned with the occasion of the first review: please ask a native English speaker to help you with the scientific writing of your paper. In this respect, see the lines 20-21,23, 35, 87, 102, 131, 254, 258, 262, 270, 308, 312-313, 348, 378, 387-388, 396, 403-404, 432, 444, 452, 456, 463-464, 467, 484, 485, 488-490, 500-501, 511-514, 520, 521, 522, 651, 652, 700. Again, in the above mentioned lines one can find plenty of phrases whose content is incomprehensible or disagreements between the subject and the predicate of sentences. Besides, the content of paragraphs between lines 650-660 and 683-693 is totally identical;
  2. On the other hand, I strongly recommend you to check the order of references as it seems that they are not numbered in the order of their occurrence in the body of the research paper. For instance, the first references can be found in line 84 and their numbers are 23 and 3. Further on, as one proceeds with the reading of your work, the bibliographical references are numbered randomly.
  3. Thirdly, I recommend you to reconsider the Conclusion section, as lines 765-772 bring no additional value in terms of presenting the output of your research, but they only repeat certain considerations that have been exposed previously in other sections of the paper.

Kind Regards!

Author Response

Thank you for your precious time reading the paper and providing detailed feedback. Regards
